# Qualified Health Claim Language affects Purchase Intentions for Green Tea Products in the United States

**DOI:** 10.3390/nu11040921

**Published:** 2019-04-24

**Authors:** Amanda Berhaupt-Glickstein, Neal H. Hooker, William K. Hallman

**Affiliations:** 1Department of Nutritional Sciences, Rutgers, The State University of New Jersey, New Brunswick, NJ 08901-2882, USA; 2John Glenn College of Public Affairs, The Ohio State University, Columbus, OH 43210, USA; hooker.27@osu.edu; 3Department of Human Ecology, Rutgers, The State University of New Jersey, New Brunswick, NJ 08901-8520, USA; hallman@aesop.rutgers.edu

**Keywords:** qualified health claim, older adult, green tea, cancer, purchase intentions

## Abstract

Qualified health claims (QHC) describe diet–disease relationships and summarize the quality and strength of evidence for a claim. Companies assert that QHCs increase sales and take legal action to ensure claims reflect their interests. Yet, there is no empirical evidence that QHCs influence consumers. Using green tea as a case study, this study investigated the effects of QHCs on purchase intentions among adults 55 years and older living in the US. An online survey using a between-subjects design examined QHCs about the relationship between green tea and the reduced risk of breast and/or prostate cancer or yukichi fruit juice and the reduced risk of gastrocoridalis, a fictitious relationship. QHCs written by a green tea company generated greater perceptions of evidence for the relationship, greater confidence in green tea and cancer, and increased purchase intentions for green tea than other QHCs. Factors that mitigated the claim’s effects on purchase intentions are: Race/ethnicity; age; importance of health claims; supplement use; health; worry about health/becoming sick with cancer; worry that led to dietary change; green tea consumption; and familiarity with the green tea–cancer. Consumers who made health-related dietary change in the past year and consider health claims important indicated greater purchase intentions than others.

## 1. Introduction

The food and dietary supplement industries have a vested interest in qualified health claims (QHCs) [1,2] because they offer a way to communicate with consumers about a product’s potential health benefit. QHCs describe the relationship between the consumption of a dietary substance and the reduced risk for a disease or health condition (i.e., diet–disease relationship). However, these claims are nuanced in that they also characterize the level of scientific support for the diet–disease relationship through a disclaimer [3]. One example of a QHC about calcium and colon or rectal polyps reads, “Very limited and preliminary evidence suggests that calcium supplements may reduce the risk of colon/rectal polyps. FDA concludes that there is little scientific evidence to support this claim” [4].

The level of evidence for a diet–disease relationship (and QHC) is determined by the US Food and Drug Administration (FDA). The FDA catalogues the scientific support for diet–disease relationships into four levels. A claim supported by strong scientific evidence (i.e., significant scientific agreement (SSA)) is described as level A. Such claims are straightforward and, importantly, do not require a disclaimer. An example of an A-level claim is: Adequate calcium throughout life, as part of a well-balanced diet, may reduce the risk of osteoporosis in later life [5]. By contrast, diet–disease relationships behind B, C, and D-level QHCs are supported by partial or incomplete evidence and require disclaimers that summarize this evidence. However, disclaimers do not indicate a level of evidence using letter-grades. Instead, the FDA prescribes the language in QHCs to accurately describe the level of scientific support [6].

Fundamentally, the value of a QHC for companies is the degree to which it persuades consumers to purchase their products [1,2]. Consequently, the disclaimer, which describes the evidence for the claimed relationship, is the crucial component of a QHC in determining product appeal to consumers. For examples of disclaimers, please see the QHCs composed by the US FDA and the court in Table 1. The disclaimer is the last sentence of the QHC.

### 1.1. Consumer Research about Qualified Health Claims

The food and supplement industries understand that consumers are motivated to buy products with health benefits and expect that QHCs will lead to increased purchases of their products. However, the expectation that QHCs can increase product purchases is based on research about the marketing impacts of A-level health claims that do not require a disclaimer. Such studies demonstrate a financial advantage [6], including greater purchase intentions [7] and increased market share [8,9]. Consumers also indicate a willingness to pay 50% to 200% more for products making health claims than for comparable products [10], including yogurt [11].

Anecdotally, companies have reported QHCs as valuable for increasing product sales [1]. Yet, only one study has examined the market potential for products with a QHC which demonstrated an increase in purchase intentions [12]. However, the experiment tested a graphic QHC, which is a format not permitted by the FDA. Therefore, there is no evidence that QHCs affect consumers’ purchase intentions or hold commercial value.

Other research has explored consumers’ understanding of the scientific support described in the QHC disclaimer. Most US consumers misunderstand QHCs as an indication of the quality of a product versus an indication of the level of evidence for a claim [13]. Further, consumers’ understanding of claims largely stems from prior experience and knowledge of a diet–disease relationship rather than a claim itself [13]. Consumers require more information for lesser-known diet–disease relationships. One study found that more detailed messages had stronger impacts on consumers when the nutrient or diet–disease link was less familiar [14]. Research also indicates that US consumers have a limited ability to act on scientific information [15] and rarely read labels because of excessive information and a lack of time [16], and these reasons may be why QHCs are infrequently found on labels [17,18].

Nonetheless, the potential of QHCs to increase product sales has motivated companies to take legal action against the FDA about how to fairly and accurately characterize the level of evidence for diet–disease relationships [2]. This is richly illustrated by the history of lawsuits filed against the FDA by manufacturers and marketers, which have sought to alter the wording of QHC disclaimers [13].

### 1.2. The Evolving Qualified Health Claim for the Green Tea and Cancer Relationship

In 2004, the FDA assigned the green tea–cancer relationship the lowest level of evidence (i.e., D grade), yet there have been seven QHCs for this claimed relationship [4,19]. Six of the seven claims are no longer in use because they mischaracterized the scientific evidence for the relationship [20,21]. The seven claims are presented in Table 1 and identify the language contested by a green tea company (Fleminger, Inc. (i.e., Fleminger)) and the FDA. QHCs are referenced by the year in which they appeared (e.g., 2004).

Of the seven claims, three were written by Fleminger (Table 1, 2004; 2008; 2010) and serve to highlight the health value of their products [1,2]. However, Fleminger’s claims were rejected by the FDA since they are scientifically inaccurate. While the three additional QHCs were written by the FDA (Table 1, 2005p; 2005b; 2011) and are scientifically accurate [22], the courts found them too technical (Table 1, 2005p; 2005b; 2011). The federal court wrote the seventh claim, which the FDA currently allows companies to use on green tea labels (Table 1, 2012) [21]. The court’s QHC aims to strike a balance between the interests of stakeholders to communicate a health benefit of a product and to ensure the accuracy of scientific evidence to prevent consumers from being misled [2].

The green tea QHC is an example that demonstrates the challenges to regulating QHCs. Using green tea as a case study, we explore the practical use of QHCs to influence consumer choice in the US. We examine the seven claims to understand their potential to influence the purchase intentions of older adults for green tea products and consider theory-based consumer and product-specific factors [23]. We also clarify whether the current QHC, permitted by the FDA, increases consumer purchase intentions for green tea products in comparison with claims no longer in use.

## 2. Materials and Methods

### 2.1. Sample

A sample of non-institutionalized, English-speaking adults aged 55 years and older, who reside in the United States (US), participated in the study. This segment was selected because older adults are affected by cancer and other chronic diseases outlined in QHCs [16,24,25]. They also represent a large portion of the population with robust purchasing power [26,27]. Research shows that older consumers are also more health-oriented than younger adults, more knowledgeable about diet and health, and adopt preventive behaviors [28], such as reading food labels [29,30] and taking dietary supplements [31,32]. One study in Finland identified health-seeking older consumers willing to purchase functional foods and believe they can influence their health through diet and lifestyle [33]. Functional foods are “foods or dietary components that may provide a health benefit beyond basic nutrition and may play a role in reducing or minimizing the risk of certain diseases and other health conditions” [34].

### 2.2. Study Design

An online survey was administered by GfK Custom Research, LLC (GfK) in January 2014. Probability-based recruitment was used to obtain a representative sample of older adults in the US. All subjects gave their informed consent. The study (protocol #12-454) was approved by the Rutgers, The State University of New Jersey Institutional Review Board.

Evidence indicates that experience with a product or prior knowledge about a diet–disease relationship motivates consumer behavior [35,36,37,38] and generates a greater perception of control over disease risk [39]. To mitigate the potential interaction of beliefs or behaviors associated with green tea and cancer, we employed a 2 × 7 between-subjects study design. This design allowed investigators to determine the effects of the QHC language. Participants were randomized into one of two conditions: (1) Green tea–breast/prostate cancer, or (2) yukichi fruit juice–gastrocoridalis, a fictitious but comparable diet–disease relationship. Yukichi fruit juice was described as a typical drink sold in stores and gastrocoridalis was introduced as a potentially painful and fatal disease. Participants were then randomized into one of seven groups so that each group viewed one QHC (see Table 1). The QHCs differed by condition in that (a) yukichi fruit juice substituted green tea as the dietary component, and (b) gastrocoridalis replaced breast and/or prostate cancer as the disease outcome.

This was a text-only study to examine consumer perceptions in response to the language in each claim, without distraction of label or package images. The QHC stimuli remained on screen for participants to refer to while responding to questions.

The survey began with, “This survey is designed to determine what Americans know and think about the health benefits of certain foods and dietary supplements. We are interested in what you currently know and feel, without using the internet or other resources to learn more about them. We are asking that you finish this survey in one sitting.” The first section asked participants about their perceptions, beliefs, and behaviors related to food and health. This included questions about their consumption of green tea and yukichi fruit juice in the past year, and their reasons for consuming them. Participants also indicated the extent to which they worry about their health and whether their worry led them to change their diet. They also indicated the extent to which they worry about becoming sick with cancer or gastrocoridalis, and whether they had a past diagnosis of either disease. Section two presented one QHC and participants responded to questions about their perceptions, based on the QHC and its description of evidence for the claimed relationship.

### 2.3. Measures

Demographic data were collected prior to the survey as part of the GfK panelist database. The modifying factors identified in the health belief model (HBM) include: Age, sex, race/ethnicity, education, employment, and income, which were also included in analysis [12,23,38,40,41,42,43,44].

Dependent variables were identified based on stakeholder interests and from the Health Claims Framework (HCF), which postulates factors that predict purchase of products with health claims [37]. Purchase intentions for green tea or yukichi fruit juice were included since they are the primary outcome for the food and supplement industries [38,41] (Table 2).

The primary outcome for the FDA is consumer understanding of the level of scientific evidence [39] described in the QHC disclaimer. To test whether the QHC language led to different perceptions of evidence, participants responded a 13-point scale (Table 3) that provided a wide range of ordinal responses for the level of evidence. Further, during analysis, the scale could be separated into four evidence levels; 0 represented “No evidence”; 1–3 represented a “D”; 4–6 a “C”; 7–9 a “B”; and 10–12 an “A”. The scale does not necessarily reflect FDA’s rating of evidence; however, it is based on the Murphy (2005) approach to develop a scale that may correspond to the levels of evidence [45]. The green tea and yukichi fruit juice claims represent a D-level of scientific evidence and are based on the same evidence.

Two questions explored the perceived benefits of green tea and yukichi fruit juice. One question inquired about their perceived disease risk reduction and the second asked about confidence in the drink as a means to reduce their risk (Table 2). Participants also indicated the perceived dose needed to achieve the health benefit (Table 2), since it is dose-dependent and in the context of the overall diet [46].

Independent variables that included product-specific and consumer-specific measures were also obtained from the HBM and HCF. Familiarity lends itself to an increase in acceptance of functional foods that bear health claims [12,38,42,47,48,49]. Therefore, familiarity with the diet–disease relationship as well as with the QHC were measured. Similarly, existing behavior may also influence purchase intentions [50]. To account for past green tea or yukichi fruit juice consumption, participants indicated the frequency of their consumption in the past year. Taste preference was also measured, since it is a prominent consideration when selecting products [38,51,52] and, for some, outweighs the perceived health value of a product [53]. Participants also reported their health status, health worries, and their perceived susceptibility for cancer or gastrocoridalis. Further, they indicated whether their worries had prompted them to make a dietary change in the past year. Other questions measured personal relevance of cancer or gastrocoridalis, knowledge about diet and health, dietary supplement use, and perceived importance of health claims when making a purchase decision.

### 2.4. Data Analysis

Descriptive statistics characterized the study sample and Spearman rank correlation tests identified associations. Based on the HCF, the measures considered to influence purchase intentions are: Perceived evidence, risk reduction, and confidence in the claimed relationship. The 13-point scale of evidence was collapsed to correspond with the grades of evidence from the FDA ranking system, where 0 corresponded to “No evidence”; 1–3 a “D”; 4–6 a “C”; 7–9 a “B”; and 10–12 an “A” (Table 3) [45]. A multivariate analysis of variance test measured interactions between these dependent variables in the green tea/yukichi fruit juice conditions and QHC groups, followed by univariate tests. Independent t-tests and Chi-square tests examined group and condition differences in terms of: Health status, worry about health, worry about cancer, worry that led to dietary change, dietary supplement use, importance of health statements on food and dietary supplement package labels, previous diagnosis of cancer, and sociodemographics.

Hierarchical multiple linear regression was used to determine the predictive value of the independent measures on purchase intentions for green tea in the presence of a QHC. The analysis focused on the green tea condition since the yukichi fruit juice–gastrocoridalis condition was fictitious. Participants in the green tea condition reported their enjoyment (or not) of the taste of green tea, whether they drank green tea in the past year, and their familiarity with the green tea–cancer relationship and with the actual claim statement. One-way analysis of variance (ANOVA) was used to compare responses between QHC groups. The Welch’s F-test results are reported when the assumption of homogeneity of variances is violated. For non-normal distributions, the potential bias was corrected with the resampling method of bootstrapping. To create a parsimonious predictive model for purchase intentions for green tea [54], nonsignificant variables were withheld from entry into the final regression model analysis, and significant predictors were added as blocks to examine their contribution to explained variance.

The HCF and HBM guided the addition of specific variables to the hierarchical model in discrete steps. At step one were the demographic, consumer-specific, and perceived susceptibility measures, step two was existing green tea consumption, step three was familiarity with the green tea–cancer relationship, and step four were the QHCs which were entered as a block of dummy-coded variables. This was done to understand how the claim statements may contribute to purchase intentions for green tea. A second regression model was created to isolate the effects of the dependent measures (confidence, risk reduction, evidence perceptions, purchase intentions) from the QHCs and included steps one through three with the removal of the dummy-coded QHCs. Adjusted R^2^ is reported to account for the number of predictor variables, and change in R^2^ is reported to indicate the contribution of each variable to the predictive model. *p* < 0.05 was considered statistically significant. Statistical analyses were performed using the Statistical Package for Social Sciences, version 22 (SPSS Inc., Chicago, IL, USA).

## 3. Results

A total of 1335 participants completed the online experiment, of the 2219 sampled, for a 60% completion rate. The experiment consisted of two conditions (green tea–cancer/yukichi fruit juice–gastrocordialis) and seven QHC groups. Each group viewed one QHC in one of two conditions; seven QHCs were about the green tea–cancer relationship (*n* = 669) and the remaining seven were identically worded QHCs about the yukichi fruit juice–gastrocoridalis relationship (*n* = 666).

The majority of participants were between the ages of 55 and 74 years old (*n* = 1123, 89.1%), White (*n* = 1060, 79.4%), who held a high school degree or more (*n* = 1233, 92.4%), with a household income under $100,000 (*n* =1028, 77.0%). Given the random assignment of participants, there were no differences between conditions or across groups by race and ethnicity, age, education, employment, household income or incidence of breast or prostate cancer (Table 4).

Overall, the majority of participants reported that they were in good health (*n* = 1010, 76%) and had never received a cancer diagnosis from their doctor (*n* =1132, 84.8%). Participants were “a little” or “somewhat” worried about their health overall (*n* = 929, 69.9%) and fewer worried about becoming sick with cancer (“somewhat” *n* = 441, 39.2%; “not at all” n = 296, 26.3%). However, most had made a dietary change in the past year (*n* = 1,022, 76.8%) due to a health-related concern. Slightly more than half of the participants reported that they drink green tea (*n* = 691, 51.8%), and most claimed to drink it because they enjoy its taste (*n* = 215, 62.5%). Of the respondents who reportedly consumed green tea in the past year (*n* = 345), fewer than 10% drank it to reduce their risk of cancer (*n* = 32, 9.3%). The majority of the participants believe they are informed about diet and health (i.e., perceived nutrition knowledge) (*n* = 1298, 97.2%) and consider health claims important on food and dietary supplement product labels (*n* = 1168, 88.0%; *n* = 926, 91.5%, respectively). No differences were found between groups or conditions with respect to general health, perceived nutrition knowledge, green tea consumption in the past year, worry about cancer, or health-related dietary changes.

Overall, participants rated the level of evidence as a “2” (i.e., D-grade) in both conditions (Table 3), and when the 13-point scale of evidence was collapsed into four grades of evidence, the results are consistent with a D level of evidence [53]. More than half of the participants were not at all confident in the ability of green tea/yukichi fruit juice to reduce the risk of cancer/gastrocoridalis (*n* = 709, 53.5%) (Mean (M)= 1.83, Standard Deviation (SD)= 1.10) but in both conditions, nearly two-thirds (*n* = 757, 57%) think there would be at least a slight risk reduction for cancer/gastrocoridalis (M = 1.98, SD = 1.10) if they drank the beverage. Most also think they would need to drink green tea/yukichi fruit juice once a week or more (*n* = 601, 80%) to reduce their risk. However, most would not buy green tea/yukichi fruit juice with the QHC (*n* = 755, 57.1%).

The majority of participants in the green tea condition reported that they were familiar with the green tea–cancer relationship (*n* = 343, 51.9%) but had never seen the QHC on a label of a green tea product (*n* = 480, 73.5%), in an advertisement or on a website or in an article (*n* = 453, 70.9%).

### 3.1. Purchase Intentions for Green Tea and Yukichi Fruit Juice

The diet–disease condition had a significant effect on perceptions of confidence, evidence, risk reduction, and purchase intentions (Pillai’s Trace = 0.102, F = 36.896, df = 4, 1296, *p* < 0.0001). A statistically significant effect was also found by QHC group on perceptions of confidence, evidence, risk reduction, and purchase intentions (Pillai’s Trace = 0.138, F = 7.719, df = 24, 5196, *p* < 0.0001) as well as a significant interaction between the condition and group on the same variables (Pillai’s Trace = 0.034, F = 1.834, df = 24, 5196, *p* < 0.0001).

However, these measures are strongly correlated (Table 5). A significant main effect of condition demonstrated that participants had greater purchase intentions for green tea than yukichi fruit juice *F*(1, 1299) = 132.320, *p* < 0.0001. There was also a significant main effect of QHC, such that claims written by Fleminger produced greater purchase intentions than claims written by the FDA, *F*(6, 1299) = 8.047, *p* < 0.0001. However, no interaction was found between condition and group, *F*(6, 1299) = 1.713, *p* = 0.114.

Significant differences were found between condition, with participants perceiving greater evidence for the green tea–cancer relationship than for the yukichi fruit juice–gastrocordialis relationship, F(1, 1299) = 25.407, *p* < 0.0001. There was also a significant main effect of the QHC group, with greater perceptions of evidence among claims written by Fleminger than those written by the FDA, F(6, 1,299) = 25.491, *p* < 0.0001. Irrespective of the QHC, participants reported greater evidence for the green tea–cancer relationship than for yukichi fruit juice and gastrocoridalis, suggesting there are other factors, in addition to the measures in the current study, that contribute to their perceptions. There was also a significant interaction between condition and QHC group on evidence ratings, meaning that participants who viewed a yukichi fruit juice–gastrocoridalis QHC rated the evidence lower than those who viewed a green tea–cancer QHC, regardless of the author, F(6, 1299) = 2.804, *p* = 0.10.

Similarly, there were main effects of both condition and QHC group on perceptions of disease reduction. Participants perceived greater reductions in cancer risk resulting from drinking green tea than gastrocoridalis from drinking yukichi fruit juice, F(1, 1299) = 36.369, *p* < 0.0001. Groups that viewed a Fleminger QHC also perceived greater reductions in disease risk than those who viewed a QHC written by others, F(6, 1299) = 19.263, *p* < 0.0001. A statistically significant interaction was also found between conditions and QHC groups on perceived risk reduction, F(6, 1299) = 3.085, *p* = 0.005. Participants’ perceptions of gastrocoridalis risk did not differ among yukichi fruit juice QHCs, while the perceived cancer risk was different among green tea QHCs.

Finally, participants in the green tea condition were significantly more confident in the claimed relationship than those in the yukichi fruit juice condition, F(1, 1299) = 78.922, *p* < 0.0001. A main effect was also identified between QHC groups. Participants who viewed a Fleminger claim reported greater confidence in the claimed relationship than in other groups, F(6, 1,299) = 16.919, *p* < 0.0001. There was also a statistically significant interaction between QHC group and condition. Participants’ confidence in the green tea condition was greater regardless of the QHC, whereas in the yukichi fruit juice–gastrocoridalis condition, participants were overall less confident, F(6, 1299) = 2.917, *p* = 0.008.

There were no statistically significant differences between the green tea and yukichi fruit juice conditions for dietary supplement use (*p* = 0.993), reported importance of health claims on dietary supplement or food labels (*p* =.505; *p* = 0.138, respectively), perceived nutrition knowledge (*p* = 0.837), worry about overall health (*p* = 0.979) that led to a dietary change in the past year (*p* = 0.360), worry about becoming sick with gastrocoridalis (*p* = 0.240) or cancer (*p* = 0.066), or a previous diagnosis of gastrocoridalis (*p* = 0.800) or cancer (*p* = 0.677).

### 3.2. Predictors of Purchase Intentions for Green Tea

To determine the factors’ predictive value for purchase intentions of green tea, consumer-specific and sociodemographic variables were individually tested for the green tea condition only (*n* = 666). The significant sociodemographic predictors are age and race/ethnicity and account for 3.5% of the variation in purchase intentions for green tea, *F*(5, 656) = 5.857, *p* < 0.0001. Participants reported lower purchase intentions with each incremental increase in age, *b* = −0.090, *t*(661) = −2.334, *p* < 0.05. Black (*b* = 0.126, *t*(656) = 3.274, *p* = 0.001). Hispanic (*b* = 0.141, *t*(656) = 3.670, *p* < 0.0001) participants reported statistically significantly greater purchase intentions for green tea than White participants.

The predictive consumer specific variables accounted for 4.1% of the variance in purchase intentions and included: Dietary supplement use and the perceived importance of health claims on food products and on dietary supplement products, *F*(3, 501) = 8.162, *p* < 0.0001. For each incremental increase in dietary supplement use in the past year, participants reported that they are *less* likely to intend to purchase green tea, (*b* = −0.114, *t*(503) = −2.582, *p* = 0.01). Yet, the more important participants consider health statements on dietary supplement products, the greater their purchase intentions (*b* = 0.135, *t*(501) = 2.209, *p* < 0.05).

Measures of perceived susceptibility that significantly predict green tea purchase intentions are: General health, worry about overall health, worry about becoming ill with cancer, and worry that led to a dietary change in the past year, *F*(4, 548) = 8.876, *p* < 0.0001. The better participants rate their general health, the greater their purchase intentions for green tea, *b* = 0.126, *t*(548) = 2.722, *p* < 0.0001. Similarly, participants who are more worried about their overall health (*b* = 0.105, *t*(550) = 2.228) or about becoming sick with cancer (*b* = 0.103, *t*(549) = 2.262) demonstrated greater purchase intentions, *p* < 0.05. Participants have greater purchase intentions with each incremental increase in dietary changes made in response to a health worry in the past year, *b* = 0.212, *t*(548) = 4.423, *p* < 0.0001. Non-explanatory predictors were removed from further analysis; sex, education, employment, income, perceived nutrition knowledge, and past cancer diagnosis.

### 3.3. Model 1

A block of the modifying sociodemographic, consumer-specific, and perceived susceptibility predictors was entered as independent variables into a regression model to predict green tea purchase intentions. These variables included: Race/ethnicity, age, perceived importance of health claims on food products and on dietary supplement products, dietary supplement use, and self-reported health status, worry about overall health, worry about becoming ill with cancer, and worry that led to a dietary change in the past year. The model significantly predicted purchase intentions for green tea and accounted for 9.3% of the variance, F(12, 406) = 4.581, *p* < 0.0001 (Table 6).

Participants who drank green tea in the past year indicated significantly greater purchase intentions (*M* = 2.73, *SD* = 1.49) than those who did not, *Welch’s F*(1, 640.890) = 87.810, *p* < 0.001. By contrast, participants who had not consumed green tea in the past year indicated that they are only “slightly likely” (*M* = 1.76, *SD* = 1.17) to buy it in the future, despite having seen a QHC about the green tea–cancer relationship. As a result, current consumption was added to the model at step 2, with a resulting significant R^2^ change = 0.052, *F*(13, 403) = 6.404, *p* < 0.0001, adj. R^2^ = 0.144 (Table 6).

There were no differences between consumers who drank green tea in the past year and those who did not in terms of whether they had seen the claim on a label, *Welch’s F*(2,114.967) = 2.826, *p* = 0.063, website, advertisement, or article, *Welch’s F*(2, 148.997) = 2.270, *p* = 0.107. Yet those who drank green tea are significantly more familiar with the green tea–cancer relationship (M = 2.11, SD = 1.10) than those who did not (M = 1.61, SD = 0.87), *Welch’s F*(1, 642.179) = 42.389, *p* < 0.0001. Accordingly, familiarity with the green tea–cancer relationship was added to the model at step 3. Results showed that familiarity with the green tea and cancer is a significant predictor of purchase intentions for green tea, *F*(14, 401) = 7.685, *p* < 0.0001, and accounted for 18.4% of the variance in the model, R^2^ change = 0.041 (Table 6). At step 4, we then entered the QHCs, which were significant predictors of future behavior, *F*(20, 395) = 6.263, *p* < 0.0001, though they only accounted for a small amount of variance in the model, adj. R^2^ = 0.202, R^2^ change = 0.029 (Table 6). In the full model, the significant predictors of purchase intentions for green tea are: The perceived importance of health statements on dietary supplements, worry about health that led to dietary change in the past year, having consumed green tea in the past year, familiarity with the diet–disease relationship, and exposure to a QHC written by Fleminger. Dietary supplement use is negatively related to purchase intentions for green tea.

### 3.4. Model 2

To understand if perceptions of evidence for the green tea–cancer relationship, risk reduction for cancer, and confidence in the claimed relationship predict purchase intentions for green tea, three additional steps were added to the model: Ratings for evidence, risk reduction, and confidence in the green tea–cancer relationship. Next, a second model was created to isolate the effects of these variables from the QHCs and included steps 1 (i.e., sociodemographics, consumer-specific, and perceived susceptibility variables), 2 (i.e., past behavior), and 3 (i.e., familiarity), and the dummy-coded QHCs were removed. The following analysis represents steps 4, 5, and 6 for the second model.

Perception of evidence was found to be a strong and significant predictor of purchase intentions for green tea, F(15, 399) = 19.781, *p* < 0.0001, adj. R^2^ = 0.315, R^2^ change = 0.128, *p* < 0.0001. Confidence in the green tea–cancer relationship is also a significantly strong predictor of future intentions to purchase green tea, F(16, 398) = 16.345, *p* < 0.0001, adj. R^2^ = 0.372, R^2^ change = 0.056, *p* < 0.0001. Entry of the third variable, perception of cancer risk reduction, showed that the overall model was significant for predicting purchase intentions for green tea, F(1, 400) = 18.736, *p* < 0.0001, although the variable did not add any more to the model, adj. R^2^ = 0.371, R^2^ change = 0.000, *p* = 0.896 (Table 6).

However, evidence, risk, and confidence are multicollinear and demonstrate strong positive correlations (Table 5), which likely explains the weak predictive value of perceived risk reduction on purchase intentions. Therefore, a backwards stepwise regression model tested the three predictors in a block. The model removed risk reduction and maintained evidence perceptions and confidence in the green tea–cancer relationship, indicating that perceived disease risk reduction did not account for any additional variance and so was removed.

The two significant predictors were re-entered into the model (i.e., perceived evidence, confidence) at Step 4. As expected, this block was a strong predictor of purchase intentions, *F*(16, 398) = 16.345, *p* < 0.0001, adj. R^2^ = 0.372, R^2^ change = 0.185 (Table 6). For every incremental increase in perceived evidence, there was a 0.130 increase in purchase intention for green tea, and for every incremental increase in confidence in the green tea–cancer relationship, there was a 0.369 increase in purchase intention. While dietary supplement use remained negatively related to purchase intentions, the importance of health statements on supplements, worry that led to dietary change, past green tea consumption, and familiarity with the claimed relationship were predictive of purchase intention (Table 6).

Neither dose nor taste were tested in a predictive model, since no association was detected between perceived dose necessary to achieve risk reductions and purchase intentions, *χ*^2^(24) = 29.845, *p* = 0.190 or with the taste of green tea, χ^2^(6) = 7.284, *p* = 0.295.

To summarize, the significant individual predictors for purchase intentions were entered into the models at step (1) and include: Sociodemographics (race/ethnicity, age); consumer-specific variables (dietary supplement use, importance of health claims on food and supplement labels); and perceived susceptibility (general health, worry about health, worry about cancer, worry that led a dietary change). Model 1 significantly predicted purchase intentions and included steps (2) green tea consumption, (3) familiarity with green tea and cancer relationship, and (4) the seven QHCs, *F*(16, 400) = 7.841, *p* < 0.0001, adj. R^2^ = 0.208, R^2^ change = 0.028. Model 2 included steps (1–3), removed step 4 (i.e., QHCs from Model 1) and replaced it with (4) evidence perceptions and (5) confidence in the green tea–cancer relationship, which led to a statistically significant increase in R^2^ of 0.185, *F*(13, 401) = 20.225, *p* < 0.0001, adj. R^2^ = 0.376.

## 4. Discussion

With respect to their future intentions to purchase green tea, the data suggest that older consumers are influenced by their past behaviors and preferences. However, even after controlling for those preferences and experiences, exposure to the claims written by Fleminger in 2004, 2008, and 2010 demonstrated greater intentions to purchase green tea than other claims. Our results also demonstrated that some QHCs increase consumer perceptions of evidence for the diet–disease relationship, which affects their confidence in the relationship and purchase intentions for green tea.

Research has identified several sociodemographic characteristics that are associated with greater acceptance of functional foods with health claims. We expected some overlap between these same characteristics of consumers who are receptive to health claims, and QHCs.

In the current study, race or ethnicity was a strong predictor of purchase intentions for green tea when entered at step one in the regression model. Black and Hispanic consumers intended to purchase green tea more than White consumers. This finding is in contrast to previous research, which has pointed to White adults as more accepting of functional food products [40,42,43,44,55] although this difference may be explained in part by the presence of a QHC which is unique to the current study. However, race or ethnicity did not remain a significant predictor of purchase intentions for green tea in the full regression model(s).

Other research has indicated that women are more receptive to functional foods and to nutrition information in health claims [37,42]. However, the current study did not find sex to significantly predict purchase intentions for green tea. Further, no other sociodemographic variables demonstrated predictive value including education, employment, or income.

Previous research also demonstrates mixed results for age as a predictor of functional food acceptance with health claims [40,42,43,44]. For example, one study found that functional foods were accepted by women 35–54 years old [42], whereas another study identified a greater range of accepting adults, 30–70 years old [41]. Still, two other studies did not find any relationship between age and functional food acceptance [42,44].

Our study initially indicated age as a strong predictor for green tea purchase intentions in adults aged 55–64 but not for those over 65 years old. However, in the full model, age was not a significant predictor of purchase intentions. More than half of our sample already drinks green tea for the taste, not to reduce their risk of cancer. While there is strong evidence that taste is a predictor for purchase intentions and behavior related to health claims [38,52], it was not correlated with purchase intentions for green tea after viewing a QHC.

Purchase intentions for yukichi fruit juice were significantly lower than for green tea. Participants could not have had prior experience with yukichi fruit juice or gastrocordialis due to their fictitious or novel nature. Therefore, consumers considered other information when responding to questions. In addition, existing green tea consumption and familiarity with the green tea–cancer relationship were both significant predictors of behavioral intentions. These findings are consistent with previous research. Familiarity and experience with a product and/or a health condition have been found to significantly predict consumer behaviors [12,38,42,47,48,49], including purchase intentions for functional foods.

Dietary supplementation was negatively associated with the intention to purchase green tea, after viewing a QHC. This is in contrast to other research that indicates consumers who regularly take supplements are more accepting of functional foods [55], such as green tea [56].

Claim language is regarded as a predictor of purchase intentions for functional products with health claims [38], and our results support this theory. While the results of the current study cannot isolate the language or wording in QHCs that may have led to greater purchase intentions for green tea, the results show that QHCs written by Fleminger led to greater purchase intentions compared with other QHCs. However, the FDA concluded that the claims written by Fleminger overstate the level of scientific support for the green tea–cancer relationship [19,21]. Our results provide support for the agency’s conclusion, demonstrating that these same QHCs produced greater perceptions of evidence and greater intentions to purchase green tea, compared with the other claims. The 2004 Fleminger QHC indicates, “There is scientific evidence supporting this health claim although the evidence is not conclusive”, while the 2008 and 2010 Fleminger claims suggest the evidence is “credible but limited”. Further, the 2010 claim identifies the FDA as the reviewer of evidence which, altogether, aimed to increase the trustworthiness of the claim. The 2005p and 2005b QHCs reference the number and quality of studies behind the claimed relationship which is similar to other diet–disease claims that the court found to be too technical for consumers to understand [20]. Although technically accurate, the claims are not consumer-friendly. Since QHCs were required by case law, it is possible that the agency sought to protect the consumer from being misled about the health value of a product by drafting overly technical claims [20,21]. The 2011 claim, which states that the FDA does not agree with the claimed relationship, provides insight into the agency’s perspective [21]. The 2012 FDA QHC suggested by the federal court as a compromise between companies’ abilities to market the health benefit of green tea and the FDA’s mission to prevent consumer confusion about scientific certainty for the relationship [21] led to greater perceptions of evidence than the FDA QHCs in 2005p/b and 2011. The 2012 QHC also produced greater purchase intentions for green tea than the FDA QHCs, but the results were not statistically significant, which suggests the current QHC may strike a balance between interests of making profit for companies and the interest of the FDA in preventing consumer confusion.

The reported confidence in the diet–disease relationship was a very strong predictor of purchase intention. On its own, the perceived extent of the reduction in the risk of cancer was associated with intentions to purchase green tea. However, it did not contribute additional predictive value to the model of purchase intentions after accounting for the measure of confidence in the green tea–cancer relationship. Our results support the hypothesis that specific claims differentially affect the perceived health benefits of green tea products since perceptions of evidence and confidence in the green tea–cancer relationship were associated with variations in the QHC.

The overall proportion of variance in purchase intentions that can be explained by the independent variables was nearly 40% (Table 6). There are several additional measures in the HCF and HBM that were not included in the current study and so, it is possible that the addition of these measures could lead to models with greater predictive power of purchase intentions for green tea with a QHC.

There is considerable potential for QHCs in the market, since most US consumers believe that functional foods can reduce the risk of becoming sick with a specific disease and believe they have some control over their personal health [57]. Participants in our sample of older consumers had greater intentions to purchase green tea if in the past year, they worried about their health and if their worry led to a dietary change. Indeed, more than three-quarters of our sample changed their diet to address health concerns. Since US consumers perceive cancer as a greater risk than other diseases, including heart disease, type 2 diabetes, and stroke [58], there is likely a considerable marketing opportunity for products associated with reducing the risks of cancer. Future research could examine the complexities of QHCs through a more interdisciplinary lens. More specifically, collaborating with market researchers could expand the application of our results to general consumer behavior and consider its place within the theory of consumption.

However, it should be noted that the current study examines QHCs for the green tea–cancer relationship, a D-grade claim, which meets the lowest level of evidence for a diet–disease relationship [19]. Future study might examine science communication in B and C grade QHCs to gain greater understanding about existing claims as well as how federal lawsuits may impact the language of QHCs, which ultimately affects consumers. Although claims do not necessarily distinguish categorically healthy versus unhealthy foods [59], the perceived health quality of products by consumers may yield different perceptions of evidence and purchase intentions [60] with QHCs.

### Strengths and Limitations

Few recent studies have examined how consumer perceptions change in response to different QHCs. This study is unique in its contribution to the understanding of the influence of QHCs on purchase intentions. Only one other study has explored purchase intentions with respect to product labels with QHCs but was limited in that the format tested has never been allowed for use on labels [12]. A major strength of this study is that it included QHCs written by different stakeholders, including Fleminger, the federal court, and the FDA, many of which have been used on labels or on the Internet.

One potential limitation of the study is that it did not include a control group that did not view any QHC. However, the existence of QHCs associated with green tea and a significant degree of prior familiarity with the green tea–cancer relationship among the participants made inclusion of a control group problematic. Instead, prior familiarity with the green tea–cancer relationship and with the QHC seen by the participant was used as a within-subjects control variable, creating a more ecologically valid design. That is, those who were unfamiliar with the relationship and had no prior familiarity with the QHC to which they were exposed serve as comparisons for those who were familiar with the relationship and/or the QHC. In addition, the study aimed to evaluate the differences in consumer perceptions among *actual* QHCs proposed or used by stakeholders and to test the assertions made by the various stakeholders about those claims. Unfortunately, members of a traditional control group would be unable to respond to the key questions about the level of perceived evidence and confidence or cancer risk reduction, based on a QHC. However, future research could test the differences between a control group and the current green tea QHC that is enforced by the FDA.

This study used a 2 × 7 between-subjects design. An alternative within-subjects study design was tested in which participants viewed several QHCs and responded to the same battery of questions. However, the time to complete the survey was considerably lengthened, threatened primacy effects, and increased participant burden, and so this design was rejected as impractical.

Finally, the study results are based on self-reported data which, in general, have a low to moderate correlation with observed behaviors [61,62]. There are likely differences between self-reported intentions and actual purchases of green tea with a QHC.

## 5. Conclusions

Overall, the QHCs impacted consumers’ perceptions of evidence for a diet–disease relationship which impacted the health perceptions for that product, as well as purchase intentions. There is a segment of older, US consumers that positively respond to green tea QHCs. Consumers who have made a dietary change to address a health concern and consider health claims on supplements to be important reported greater purchase intentions than others. Consumers who perceived there to be greater evidence for the green tea–cancer relationship and reported greater confidence in the claimed relationship also intended to purchase green tea more than others. The three claims written by Fleminger, the green tea manufacturer, led to perceptions of greater evidence, perceived cancer risk reduction, and confidence in the claimed relationship, which also produced greater purchase intentions for green tea when compared with QHCs written and permitted by the FDA or the Court.

## Figures and Tables

**Table 1 nutrients-11-00921-t001:** Seven qualified health claims petitioned, unlawfully used by manufacturers, or prescribed by the US Food and Drug Administration or Federal Court, 2004–2012 (***n*** = 1335).

Author	Year	Status	Qualified Health Claim	*n*
Fleminger	2004	Scientifically inaccurate Petitioned claim	Daily consumption of 40 ounces of typical green tea containing 170µg/mL of natural (-) epigallocatechin gallate (EGCG) may reduce the risk of certain forms of cancer. **There is scientific evidence supporting this health claim although the evidence is not conclusive**.	185
Fleminger	2008	Scientifically inaccurate Unlawfully used	Green tea may reduce the risk of cancer of the breast and the prostate. There is **credible evidence** supporting this claim although the evidence is limited.	176
Fleminger	2010	Scientifically inaccurate Illegally names FDA Unlawfully used	Green tea may reduce the risk of breast and prostate cancers. The **FDA** has concluded that there is **credible evidence** supporting this claim although the evidence is limited.	185
FDA	2005p	Scientifically accurate Overly technical No longer allowed	**One weak and limited study** does not show that drinking green tea reduces the risk of prostate cancer, but **another weak and limited study** suggests that drinking green tea may reduce this risk. Based on these studies, FDA concludes that it is **highly unlikely** that green tea reduces the risk of prostate cancer.	211
FDA	2005b	Scientifically accurate Overly technical No longer allowed	**Two studies** do not show that drinking green tea reduces the risk of breast cancer in women, but **one weaker, more limited study** suggests that drinking green tea may reduce this risk. Based on these studies, FDA concludes that it is **highly unlikely** that green tea reduces the risk of breast cancer.	179
FDA	2011	Scientifically accurate Disclaimer negates claim No longer allowed	Drinking green tea may reduce the risk of breast or prostate cancer. **FDA does not agree** that green tea may reduce that risk because there is very little scientific evidence for the claim.	206
Federal Court	2012	Scientifically accurate Technically appropriate Allowed	Green tea may reduce the risk of breast or prostate cancer. FDA has concluded that there is very little scientific evidence for this claim.	193

Note: Bold words in qualified health claims (QHCs) indicate issues in the description of evidence.

**Table 2 nutrients-11-00921-t002:** Variable descriptions.

	Variable	Survey Question	Scale
**DEPENDENT VARIABLES**	**Purchase intention**	If priced the same as other GTs *(YFJ **) without this statement, how likely is it that you would purchase a bottle of GT (YFJ) with this statement?	1 = Not at all likely, 2 = Slightly likely, 3 = Somewhat likely, 4 = Fairly likely, 5 = Very likely, 6 = Extremely likely, 7 = Absolutely certain
	**Evidence**	Based on this statement, how much evidence is there that drinking GT (YFJ) may reduce the risk of Cancer (Gastrocordialis)?	See Table 3
	**Confidence**	Based on this statement, how confident are you that drinking this GT (YFJ) will reduce the risk of Cancer (Gastrocordialis)?	1 = Not at all confident, 2 = Slightly confident, 3 = Somewhat confident, 4 = Fairly confident, 5 = Very confident, 6 = Extremely confident, 7 = Absolutely confident
	**Risk perception**	Based on this statement, how much can drinking GT (YFJ) (as part of a regular diet) reduce the risk of Cancer (Gastrocordialis)?	1 = Not at all, 2 = Slight reduction, 3 = Some reduction, 4 = Modest reduction, 5 = Large reduction, 6 = Extreme reduction, 7 = Complete reduction
	**Dose**	How often would someone have to drink GT (YFJ) to reduce their risk of Cancer (Gastrocordialis)?	1 = Never, 2 = Less than once a month, 3 = Once a month, 4 = 2–3 times a month, 5 = At least once a week
**INDEPENDENT VARIABLES**	**Reason for drinking GT (YFJ)?**	Why do you drink GT (YFJ)?	1 = To reduce the risk of X-type cancer, 2 = For another specific health reason, 3 = I enjoy the taste, 4 = For other reasons
	**Familiarity—QHC**	Have you ever seen this [QHC] on a: food or dietary supplement label/in an advertisement/on a website or in an article?	−1 = No, 0 = I don’t know, 3 = Yes
	**Familiarity—diet–disease relationship**	Whether you have seen this statement or not, how familiar are you with the idea that drinking GT (YFJ) can reduce the risk of Cancer (Gastrocordialis)?	1 = Not all familiar, 2 = Somewhat, 3 = Fairly, 4 = Very, 5 = Extremely familiar
	**Health Status**	Would you say that in general your health is:	1 = Poor, 2 = Fair, 3 = Good, 4 = Very good, 5 = Excellent
	**Cancer Worry**	How worried are you about becoming ill with Cancer (Gastrocordialis)?	1 = Not at all, 2 = Somewhat, 3 = Moderately, 4 = Very, 5 = Extremely
	**Diet Δ for Health Worry**	How much has worrying about your health led you to change the way you ate in the past year?	1 = Not at all, 2 = A little, 3 = Somewhat, 4 = Quite a bit, 5 = All the time
	**Health Worry**	How often have you worried about your overall health in the past year?	1 = Not at all, 2 = A little, 3 = Somewhat, 4 = Quite a bit, 5 = All the time
	**Cancer Diagnosis**	Has a doctor ever told you that you had Cancer (Gastrocordialis)?	0 = No, 1 = Yes
	**Perceived Nutrition Knowledge**	How well informed are you about diet and health?	1 = Not informed, 2 = Somewhat, 3 = Fairly, 4 = Very, 5 = Extremely informed
	**Existing behavior**	Over the past 12 months, how often did you drink GT (YFJ)?	1 = Never, 2 = Less than once a month, 3 = Once a month, 4 = 2–3 times a month, 5 = At least once a week
	**Importance of Health Claims on Supplement Labels**	When you consider buying a new dietary supplement, how important are statements on the label that describe the product’s health benefits?	1 = Not at all important, 2 = Somewhat, 3 = Important, 4 = Very, 5 = Absolutely essential
	**Supplement Use**	Over the past 12 months, how often did you take a dietary supplement? A dietary supplement could be a vitamin, or a mineral that is taken to supplement the diet (e.g., MVI).	1 = Never, 2 = Less than 1 day/month, 3 = 1–3 days/month, 4 = 1–3 days/week, 5 = 4–6 days/week, 6 = Everyday
	**Importance of Health Claims on Food Labels**	When you consider buying a new food product, how important are statements on the label that describe the product’s health benefits?	1 = Not at all important, 2 = Somewhat, 3 = Important, 4 = Very, 5 = Absolutely essential

* GT: Green tea, ** YFJ: Yukichi fruit juice.

**Table 3 nutrients-11-00921-t003:** Semantic differential scale of evidence and collapsed categories for data analysis.

Evidence Scale	None	Minimal				Some				Complete			
	o	o	o	o	o	o	o	o	o	o	o	o	o
**Level of evidence ***	N/A	D			C			B			A		
**FDA claim category**	N/A	Qualified Health Claim									Health Claim		

* A footnote in the 2009 Final Guidance indicated the health claim grading system is no longer in effect (Food and Drug Administration, 2003b; Food and Drug Administration, 2011). However, an implicit scale of evidence remains, as it is the fundamental difference between health claims and qualified health claims. For this reason, the current study used the 2003 scale of evidence to explore consumer perceptions of evidence. o = response scale points.

**Table 4 nutrients-11-00921-t004:** Sample characteristics, *n* (%).

	Green Tea	Yukichi Fruit Juice	All Participants
**Characteristics**	(*n* = 666)	(*n* = 669)	(***n*** = 1335)
**Sex ***			
Female	323 (46.9)	365 (53.0)	688 (51.5)
Male	343 (53.0)	304 (46.9)	647 (48.5)
**Race/Ethnicity**			
White	532 (39.9)	528 (39.6)	1060 (79.4)
Black	65 (4.9)	49 (3.7)	114 (8.5)
Hispanic	39 (3.0)	39 (3.0)	77 (6.0)
Other	20 (1.5)	15 (1.1)	35 (2.6)
2 or more races	26 (1.9)	23 (1.7)	49 (3.7)
**Age**			
55 to 64 years old	330 (24.7)	328 (24.6)	658 (49.3)
65 to 74 years old	229 (17.2)	236 (17.7)	465 (34.8)
75 or older	107 (8.0)	105 (7.9)	212 (15.9)
**Education**			
Less than high school	52 (7.8)	50 (7.5)	102 (7.6)
High school	244 (36.6)	209 (31.2)	453 (33.9)
Some college	195 (29.3)	197 (29.4)	392 (29.4)
Bachelor’s degree or higher	175 (26.3)	213 (31.8)	388 (29.1)
**Household Income**			
$0–$49,999	283 (21.2)	309 (23.1)	592 (44.3)
$50,000–$99,999	229 (17.2)	207 (15.5)	436 (32.7)
$100,000–$149,999	98 (7.3)	107 (8.0)	205 (15.4)
$150,000 and above	52 (3.9)	50 (3.7)	102 (7.6)
**Health Status**			
Cancer (general)	104 (7.8)	99 (7.4)	203 (15.2)
Breast cancer	19 (1.4)	28 (2.1)	47 (3.5)
Prostate cancer	25 (1.9)	20 (1.5)	45 (3.4)
Gastrocoridalis	7 (0.5)	8 (0.6)	15 (1.1)

* *p* < 0.05.

**Table 5 nutrients-11-00921-t005:** Group means and standard deviations of ratings for evidence, cancer risk reduction, and confidence in the green tea–cancer relationship by QHC. Correlations (rs) between group perceptions of evidence, risk reduction, and confidence in the green tea–cancer relationship.

							Risk Reduction	Confidence
Measure	QHC	*n*	*Min*	*Max*	*M*	*SD*	*r_s_*, *sig*	
**Evidence**	2004	95	1	10	4.18	2.67	0.536, p < 0.01	0.523, p < 0.01
	2008	93	1	13	3.91	2.69		
	2010	102	1	13	4.70	2.95		
	2005P	92	1	11	2.43	1.98		
	2005B	88	1	12	2.72	2.18		
	2011	96	1	8	2.38	1.77		
	2012	94	1	12	3.23	2.62		
**Risk reduction**	2004	95	1	5	2.65	1.07		0.804, p < 0.01
	2008	95	1	6	2.35	1.12		
	2010	102	1	5	2.59	1.18		
	2005P	93	1	5	1.81	1.09		
	2005B	90	1	5	1.73	0.88		
	2011	96	1	4	1.98	0.93		
	2012	94	1	5	2.06	1.09		
**Confidence**	2004	95	1	6	2.47	1.17		
	2008	95	1	5	2.22	1.09		
	2010	102	1	6	2.55	1.25		
	2005P	93	1	7	1.71	1.09		
	2005B	90	1	7	1.73	1.12		
	2011	96	1	5	1.86	1.00		
	2012	94	1	5	2.07	1.23		

Min = Minimum rating; Max = Maximum rating; M = mean; SD = standard deviation.

**Table 6 nutrients-11-00921-t006:** Hierarchical regression analysis for variables predicting purchase intentions for green tea.

	*Model 1*					*Model 2*				
Variable				95% CI					95% CI	
	*B*	*SE_B_*	*Beta*	*Lower*	*Upper*	*B*	*SE_B_*	*Beta*	*Lower*	*Upper*
**Age**	−0.029	0.091	−0.014	−0.207	0.150	0.017	0.081	0.009	−0.142	0.176
**Race/Ethnicity**										
Black	0.154	0.265	0.027	−0.367	0.676	−0.002	0.236	0.000	−0.467	0.463
Hispanic	0.487	0.254	0.087	−0.012	0.985	0.284	0.224	0.051	−0.155	0.723
Other, non-Hispanic	0.307	0.379	0.037	−0.437	1.051	0.271	0.335	0.033	−0.388	0.930
2+, non-Hispanic	0.281	0.342	0.037	−3.90	0.953	0.258	0.302	0.034	−0.335	0.852
**DS Use**	−0.163	0.055	−0.133 **	−0.272	−0.054	−0.121	0.049	−0.099 *	−0.218	−0.024
**Importance of Health Claims**										
Food Labels	−0.053	0.082	−0.042	−0.215	0.109	−0.081	0.073	−0.064	−0.224	0.062
DS Labels	0.169	0.077	0.138 *	0.017	0.321	0.131	0.068	0.107	−0.003	0.266
**Health Status**	0.057	0.088	0.034	−0.115	0.230	0.026	0.077	0.016	−0.126	0.179
**Health Worry**	−0.034	0.089	−0.022	−0.209	0.141	−0.014	0.079	−0.010	−0.169	0.140
**Cancer Worry**	0.062	0.072	0.044	−0.080	0.205	−0.024	0.064	−0.017	−0.149	0.101
**Worry Diet ∆**	0.155	0.075	0.115 *	0.008	0.301	0.142	0.065	0.106 *	0.014	0.271
**Green Tea Consumption**	0.200	0.045	0.213 **	0.111	0.288	0.171	0.040	0.182 **	0.093	0.250
**Familiarity**	0.272	0.066	0.200 **	0.143	0.402	0.048	0.061	0.035	−0.072	0.168
**QHC Group**										
QHC 2004	0.617	0.246	0.150 *	0.133	1.101					
QHC 2008	0.609	0.250	0.140 *	0.117	1.101					
QHC 2010	0.654	0.237	0.164 **	0.188	1.121					
QHC 2005b	0.126	0.2407	0.031	−0.346	0.599					
QHC 2011	0.162	0.246	0.039	−0.320	0.645					
QHC 2012	0.358	0.241	0.088	−0.115	0.832					
**Evidence**						0.130	0.026	0.238 **	0.079	0.181
**Confidence**						0.369	0.060	0.314 **	0.250	0.487
***Adjusted R^2^***		0.202					0.372			
***F* for ∆ in *R^2^***		2.535*					60.891 **			

CI = confidence interval; *B* = unstandardized beta; *SE_B_* = standard error for the unstandardized beta; *Beta* = standardized beta; * *p* < 0.05 ** *p* < 0.01; Reference categories: Race/ethnicity = White; QHC Group = 2005 p.

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
