# Peer review of "Qualified Health Claim Language affects Purchase Intentions for Green Tea Products in the United States"

_nutrients, 2019, doi:10.3390/nu11040921_

Reviewer 1 Report

I want to congratulate to the authors for their work. It was a pleasure to me read the paper. The paper is pretty clear, well-structured and present very well the present state-of-art of health claims in the US. The methodology used by the authors is simple but with powerful results and implications. The main contribution of the paper is expanding the literature about consumers' preferences about health claims and indirectly with the fighting against non-communicable diseases. 

I only suggest to the authors check few typos. The first one, on page 2, line 54 and 82. The first phrase must be a subheading and do not part of the paragraph. The second one, page 7, line 160, abbreviators 'HBM' and 'HCF' wasn't defined before.

Author Response

Dear Reviewer 1, Thank you very much for your review and suggestions. We are pleased that you enjoyed reading our paper. We have addressed the issues that you identified, which are outlined in the table below. Thank you again.

Page

Line(s)

Reviewer Point

Author Response

Edit(s) in Manuscript

2

54

The first phrase must be   a subheading and do not part of the paragraph.

Thank you for catching   this error. We have addressed the issue with appropriate formatting.

Line 55: 1.1. Consumer research about Qualified   Health Claims.

2

82

The first phrase must be   a subheading and do not part of the paragraph.

Thank you for catching this   error. We have addressed the issue with appropriate formatting.

Line 83: 1.2. The evolving Qualified Health Claim   for the green tea and cancer relationship.

7

160

Abbreviators 'HBM' and   'HCF' wasn't defined before. 

Thank you for pointing   this out. We mistakenly did not include abbreviations in the earlier text and   have made this edit.

Line 147: The modifying   factors identified in the Health Belief Model (HBM) include…

Line 149-50: Dependent   variables were identified based on stakeholder interests and from the Health   Claims Framework (HCF)…

Reviewer 2 Report

Nutrients-487122  Qualified Health Claim Language affects Purchase Intentions for Green
Tea Products in the United States

General comment: This study was aimed at exploring how the purchase intent of a sample of older U.S. consumers was influenced by the content of qualified health claims on the green tea – breast/prostate cancer relationship and a fictitious yyukichi fruit juice - of gastrocoridalis.  The empirical data were collected from an online experiment was administered to a convenience sample of 1,335 U.S. consumers 55 years or older.  In my view, the study provides some new and potentially relevant information on purchase intent toward products that might consider carrying a qualified health claim with very weak scientific evidence.  Yet, the value of the manuscript has been severely diminished by many inaccurate and confusing statements and statements that lack justification, which in turn raise significant concern in my mind about the overall trustworthiness of the manuscript.  Below are examples of such statements and some other comments.

Inaccurate and confusing  statements and statements that must be justified

Table 3, Lines 175-176, Line 231: where is the justification for the “correspondence” between the 0-to-12 scale and FDA’s QHC classification?  Unless the correspondence has been used by the FDA or established in the scientific literature, it is misleading to present it in the study because it would create the erroneous impression that such a correspondence does exist.  Furthermore, FDA’s review of QHC evidence is based on many aspects of evidence, e.g., quantity, quality, nature of study.  But the labels used in the study, minimal, some, and complete do not accurately characterize the FDA algorithm.

Table 6, Model 1: why is the FDA prostate QHC chosen as the reference claim?  This choice may have significant implications for the interpretation of the study because, unlike other claims examined, both 2005 P and 2005 B claims mention only one cancer rather than both cancers.  By using the 2005 P claim as the reference claim, the comparison may have been affected by not only the language but also the content of claims.  And, it is not clear if there is a way to disentangle between the two effects.  Hence, the reference claim raises question about the validity of the most important findings.      

Lines 190-193: it is difficult to understand the analytic strategy used.  The paper states here “non-significant variables were withheld from entry into the final regression model analysis and significant predictors were added as blocks to examine their contribution to explained variance.”  Yet, why are non-significant variables still included in the “final” models, e.g., Models 1 and 2 in Table 6.  Furthermore, both the Results and the Summary sections discuss findings on “non-significant” variables and findings that are significant at the early steps of modeling but become not significant at the final steps, e.g., Lines 373-376.  The existence of such confusion appears to make the manuscript exceedingly and unnecessarily difficult to follow.

Table 1: what is the “N”?  (1) If the total sample (1,335) was split into two product groups (green tea 666 and yyukichi fruit juice 669) and each study participant saw only one product, then the “green tea” sample could not have the sample size as stated in the table. (2) The source of this footnote must be identified and referenced: “*As deemed by the US Food and Drug Administration.”

Line 45: the correct claim language ends with “in later life,” not “later in life.”

Line 61: it does not appear that Ref 11 and 12 investigated any “A level” claims.

Lines 91 and 93: Who “rejected” the claim(s) must be identified.

Table 2: are the labels for the two “familiarity” variables misplaced?

Throughout the paper: the term “functional food” is often used and there appears to be a presumption that both studied products are functional food.  Yet, the term is never defined and there is no discussion of why either product is considered a functional food.

Line 200: are these “dependent” measures not “independent” measures?

Line 224: where does the 9.3% come from?  What is the denominator?

Other comments

Line 51: suggest adding one or two examples of a disclaimer.

Table 1: suggest clarifying somewhere which claims are petitioned and unlawfully used.  

Lines 103-104: cannot find where this is discussed in the Results and/or Summary section: “We also clarify whether the current QHC, permitted by the FDA, increases consumer purchase intentions for green tea products in comparison with claims no longer in use.”

Line 182: Is a linear regression (assuming it means an ordinary least squares regression) the appropriate statistical model for dependent variables in this study?  All dependent variables are on a discrete and ordered measure.  Could alternative models such as an ordered logistic/probit model better match the dependent variables?

Lines 207-211: suggest adding information on the response rate or the cooperation rate, i.e., how many panelists were invited to participate?

Table 4: note that percentages are in parentheses.

Lines 262 and 416: explain what the “other factors” are or may be.

Line 281, why is this finding “as expected”?

Table 6: what do * and ** denote?

Line 428: could an attempt be made to explain how the language of a claim could make a difference in purchase intent by speculating the focus of the various claims, i.e., how they describe the strength of the “evidence” and how their disclaim language differs?     

Author Response

Dear Reviewer 2, Thank you very much for your detailed review. We addressed your concerns throughout the paper. Please see the table below with a detailed response to your inquiries and suggestions. Thank you again for your diligence.

Page

Table

Line(s)

Reviewer Point

Author Response

Edit(s) in Manuscript

3

175-6;231

where is the   justification for the “correspondence” between the 0-to-12 scale and FDA’s   QHC classification?  Unless the correspondence has been used by the FDA   or established in the scientific literature, it is misleading to present it   in the study because it would create the erroneous impression that such a   correspondence does exist.  Furthermore, FDA’s review of QHC evidence is   based on many aspects of evidence, e.g., quantity, quality, nature of   study.  But the labels used in the study, minimal, some, and complete do   not accurately characterize the FDA algorithm.

Thank you. Sure. We will   explain our logic behind the question and response scale per your comment. And   thank you, we’ve also added this reference to our manuscript to reflect your   query.

The 13-point scale is   based, in-part, on the rationale in Murphy, 2005 where he stated “It   therefore was necessary to establish a rationale for determining which score,   or range of scores, on the questionnaire’s seven-point certainty scale should   correspond to each of FDA’s four designated levels of scientific support.   Although the system that was adopted is somewhat arbitrary and not   necessarily superior to other possible assignment rules, it does appear   reasonable and internally consistent” (pg. 16).

We recognize that the   scale may be imperfect but to further explain our logic: we initially created   a semantic differential scale of 0 to 100, where zero = no evidence and 100 =   complete evidence. We agreed 100 points were too many and that scale   descriptors would alleviate response burden. First, we selected the scale   descriptors from existing FDA enforcement letters and avoided QHC language.   We reduced the scale to 13 points to offer respondents a middle choice and to   separate the scale into four, 3-point sections (A-D), plus 0 for no evidence.  

Lines 182-84: The   13-point evidence scale was collapsed to correspond with the grades of   evidence from the FDA ranking system where 0 corresponded to “No evidence”;   1-3 a “D”; 4-6 a “C”; 7-9 a “B”; and 10-12 an “A” (Table 3) [].

Lines 242-44: Overall,   participants rated the level of evidence as a “2” (i.e. D-grade) in both   conditions (Table 3) and when the 12-point scale of evidence was collapsed   into four grades of evidence, the results are consistent with FDA’s   evaluation of the scientific evidence42 [].

6

why is the FDA prostate   QHC chosen as the reference claim?  This choice may have significant   implications for the interpretation of the study because, unlike other claims   examined, both 2005 P and 2005 B claims mention only one cancer rather than   both cancers.  By using the 2005 P claim as the reference claim, the   comparison may have been affected by not only the language but also the   content of claims.  And, it is not clear if there is a way to   disentangle between the two effects.  Hence, the reference claim raises   question about the validity of the most important findings.

Thank you for your   inquiry. Table 1 lists the seven QHCs in this study. Three claims were   written by the manufacturer and were rejected by the FDA, so these claims are   scientifically inaccurate (i.e. 2004, 2008, 2010) and therefore, not selected   as a reference group.

Three claims were   written by the FDA (i.e. 2005p, 2005b, 2011). The 2011 QHC disclaimer [“FDA   does not agree that green tea may reduce that risk because there is very   little scientific evidence for the claim”] was found to negate the claimed   relationship by the federal court. It was therefore not selected as a   reference group. The 2005p and 2005b QHCs were written by the FDA and are   scientifically accurate. We chose 2005p instead 2005b since the mean evidence   rating was the lowest (M=2.43 – Table 5) and demonstrated the greatest   completion rate (n=211).

The 2012 claim was   written by the court and part of our research was to understand how responses   to this claim compared with scientifically accurate claims (i.e. 2005p,   2005b) and inaccurate claims (i.e. 2004, 2008, 2010, 2011).

190-3; 373-6

it is difficult to   understand the analytic strategy used.  The paper states here   “non-significant variables were withheld from entry into the final regression   model analysis and significant predictors were added as blocks to examine   their contribution to explained variance.”  Yet, why are non-significant   variables still included in the “final” models, e.g., Models 1 and 2 in Table   6.  Furthermore, both the Results and the Summary sections discuss   findings on “non-significant” variables and findings that are significant at   the early steps of modeling but become not significant at the final steps,   e.g., Lines 373-376.  The existence of such confusion appears to make   the manuscript exceedingly and unnecessarily difficult to follow.

Thank you for your   comment. The two sociodemographic variables that were significant predictors   of purchase intentions for green tea are race/ethnicity and age (Lines   362-4). Therefore, they were added to Models 1 at Step 1 in which they were   significant however they were non-significant in the full model (Lines   410-414; Table 6).

Race/ethnicity and age   were again entered as part of Step 1 in Model 2. In this model, we removed   the dummy-coded QHCs to isolate the effects of evidence, risk reduction, and   confidence. So Steps 1-3 were again added to Model 2 to understand the   effects of the dependent variables. In Model 2, the sociodemographic variables   remained non-significant (Table 6).

We understand your   concerns with the summary paragraph and have adjusted the language to increase   its clarity.

Lines 386-95: “To   summarize, the significant individual predictors for purchase intentions were   entered into the models at step (1) and include: socio-demographics   (race/ethnicity, age); consumer-specific variables (dietary supplement use,   importance of health claims on food and supplement labels); and perceived   susceptibility (general health, worry about health, worry about cancer, worry   that led a dietary change). Model 1 significantly predicted purchase   intentions and included steps (2) green tea consumption, (3) familiarity with   green tea and cancer relationship, and (4) the seven QHCs, F(16, 400) =   7.841, p < .0001, adj. R2 = .208, R2 change = .028. Model 2 included steps   (1-3), removed step 4 [i.e. QHCs from Model 1] and replaced it with (4)   evidence perceptions and (5) confidence in the green tea-cancer relationship,   which led to a statistically significant increase in R2 of .185, F(13, 401) =   20.225, p < .0001, adj. R2 = .376.”

1

what is the “N”?    (1) If the total sample (1,335) was split into two product groups (green tea   666 and yukichi fruit juice 669) and each study participant saw only one   product, then the “green tea” sample could not have the sample size as stated   in the table.

(2) The source of this   footnote must be identified and referenced: “*As deemed by the US Food and   Drug Administration.”

Thank you.

(1) This is an error.   Table 1 includes the entire sample (N=1335) however that is not clear. We   have added this to the table title.

(2) Removed

(1) Title: Table 1.   Seven qualified health claims petitioned, unlawfully used by manufacturers,   or prescribed by the US Food and Drug Administration, 2004-2012 (N=1,335).

(2)

45

the correct claim   language ends with “in later life,” not “later in life.”

Thank you. We’ve made   this change.

Lines 44-5: Adequate   calcium throughout life, as part of a well-balanced diet, may reduce the risk   of osteoporosis in later life.

61

it does not appear that   Ref 11 and 12 investigated any “A level” claims.

References 11 and 12 include   health claims but not necessarily A claims. We changed the language in the manuscript   to be more accurate. Thank you.

Lines 60-2: Consumers   also indicate a willingness to pay 50% to 200% more for products making   health claims than for comparable products [11], including yogurt [12].

91

Who “rejected” the   claim(s) must be identified.

Good point, thanks. We   indicated who or what entity rejected the claim.

Lines 91-4: However,   Fleminger’s claims were rejected by the FDA since they are scientifically   inaccurate. While the three additional QHCs were written by the FDA (Table 1,   2005p; 2005b; 2011) and are scientifically accurate [23], the courts found   them too technical (Table 1, 2005p; 2005b; 2011).

2

are the labels for the   two “familiarity” variables misplaced?

Correct! Thank you.

See Table 2 for language   switched between Familiarity variables.

All

the term “functional   food” is often used and there appears to be a presumption that both studied   products are functional food.  Yet, the term is never defined and there   is no discussion of why either product is considered a functional food.

Thank you. We have added   a definition of functional food to the manuscript to clarify the term.

Lines 113-5: Functional   foods are “foods or dietary components that may provide a health benefit   beyond basic nutrition and may play a role in reducing or minimizing the risk   of certain diseases and other health conditions” (AND, Position paper 2013).

200

are these “dependent”   measures not “independent” measures?

Thank you for your thoughtful   inquiry. The measures outlined in top portion of Table 2 are dependent since   the responses may change as a result of exposure to a QHC. The independent   measures are those factors that will not change as a result of the QHC. The   Wills et al. (2012) review and conceptual framework guided this study which   identified several variables in the same manner.

224

where does the 9.3% come   from?  What is the denominator?

Thank you. The sample   was divided into two conditions. Of the participants in the green tea   condition who drank green tea, they indicated why they chose to do so. The   denominator is smaller. We have added language to the manuscript to clarify   this result.

Lines 233-4: Of the   respondents who reportedly consumed green tea in the past year (n=345), fewer   than 10% drank it to reduce their risk of cancer (n=32, 9.3%).

51

suggest adding one or   two examples of a disclaimer.

Thank you. To maintain   the length of the manuscript, we have added language to refer to Table 1 that   includes this information.

Lines 53-4: For examples   of disclaimers, please see the QHCs composed by the US FDA and the court in   Table 1. The last sentence of these four QHCs are disclaimers. 

1

suggest clarifying   somewhere which claims are petitioned and unlawfully used.

Okay, thank you.

Please refer to Table 1,   Status column for added language.

103-4

cannot find where this   is discussed in the Results and/or Summary section: “We also clarify whether   the current QHC, permitted by the FDA, increases consumer purchase intentions   for green tea products in comparison with claims no longer in use.”

Thank you. We added some   language to address this concern.

Lines 461-4: “The 2012   QHC also produced greater purchase intentions for green tea than the FDA QHCs   but the results were not statistically significant, which suggests the   current QHC may strike the balance between interests of making profit for   companies and the interest of the FDA to prevent consumer confusion.”

182

Is a linear regression   (assuming it means an ordinary least squares regression) the appropriate   statistical model for dependent variables in this study?  All dependent   variables are on a discrete and ordered measure.  Could alternative   models such as an ordered logistic/probit model better match the dependent   variables?

Thank you. It is   possible that an ordered logistic/probit model could have  examined this data. However, the authors   agree that the results and interpretation would be far more complicated,   particularly any interaction effects, and less appropriate for our target   audience (nutrition professionals, regulators, the courts) who will have less   experience with logistic regression/probit analyses than linear regression.

207-11

suggest adding   information on the response rate or the cooperation rate, i.e., how many   panelists were invited to participate?

Thank you. We have added   the completion rate to the manuscript.

Lines 217-8: A total of   1,335 participants completed the online experiment, of the 2,219 sampled, for   a 60% completion rate.

4

note that percentages   are in parentheses.

Thank you. We added that   information to the title of Table 4.

Please refer to the   Title in Table 4.

262;416

explain what the “other   factors” are or may be.

Thank you. We have added   language to address your concern.

Lines 272-4:   Irrespective of the QHC, participants reported greater evidence for the green   tea-cancer relationship than for yukichi fruit juice and gastrocoridalis   suggesting there are other factors, in addition to the measures in the   current study, that contribute to their perceptions.

281

why is this finding “as expected”?

Thank you. We have removed   this phrase from the manuscript.

6

what do * and ** denote?

Thank you. We have added   appropriate footnotes to denote the two symbols.

Please refer to Table 6.  

428

could an attempt be made   to explain how the language of a claim could make a difference in purchase   intent by speculating the focus of the various claims, i.e., how they   describe the strength of the “evidence” and how their disclaim language   differs?    

Ok, thank you.

Lines 440-464: “Claim   language is regarded as a predictor of purchase intentions for functional   products with health claims [38] and our results support this theory. While   the results of the current study cannot isolate the language or wording in   QHCs that may have led to greater purchase intentions for green tea, the   results show that QHCs written by Fleminger led to greater purchase   intentions compared with other QHCs. However, the FDA concluded that the   claims written by Fleminger overstate the level of scientific support for the   green tea-cancer relationship [20,22]. Our results provide support for the   agency’s conclusion, demonstrating that these same QHCs produced greater   perceptions of evidence and greater intentions to purchase green tea,   compared with the other claims. The 2004 Fleminger QHC indicates, “There is   scientific evidence supporting this health claim although the evidence is not   conclusive” while the 2008 and 2010 Fleminger claims suggest the evidence is   “credible but limited,”. Further, the 2010 claim identifies the FDA as the   reviewer of evidence which altogether, aimed to increase the trustworthiness   of the claim. The 2005p and 2005b QHCs reference the number and quality of   studies behind the claimed relationship which is similar to other   diet-disease claims that the court found to be too technical for consumers to   understand [21]. Although technically accurate, the claims are not   consumer-friendly. Since QHCs were required by case law, it is possible that   the agency sought to protect the consumer from being misled about the health   value of a product by drafting overly technical claims [21-22]. The 2011   claim, which states that the FDA does not agree with the claimed   relationship, provides insight into the agency’s perspective [22]. The 2012   FDA QHC suggested by the federal court as a compromise between companies’   abilities to market the health benefit of green tea and the FDA’s mission to   prevent consumer confusion about scientific certainty for the relationship   (Fleminger, Inc. v. US DHH, 2012) led to greater perceptions of evidence than   the FDA QHCs in 2005p/b and 2011. The 2012 QHC also produced greater purchase   intentions for green tea than the FDA QHCs but the results were not   statistically significant, which suggests the current QHC may strike a   balance between interests of making profit for companies and the interest of   the FDA in preventing consumer confusion.”

Reviewer 3 Report

The article presents a very important topic, which concerns the influence of the Qualified Health Claims on consumer purchase intentions. My evaluation of the paper is high, particularly in the case of methodology, which was described in an exhaustive way. The choice of green tea and the target group is proper and justified. The results are clearly described and wide discussion has been conducted. The structure of the paper is correct as well.

Nevertheless, I have some comments that the authors may take into consideration:

1.       In my opinion, the article would have even more scientific soundness, if there were some references to the theory of consumption, particularly to the consumer behavior.

2.       The contribution would gain in value, if the importance of results for the market would be wider described. I recommend to relate the study to market investigation or carry out more interdisciplinary research in future.

3.       The abstract lacks the aim of the paper as well as research questions and/or hypotheses.

4.       I think that the sentence “Perhaps participants in our study that take (….) drink green tea” (p. 15, l. 424-426) should not appear in the paper because the participants’ beliefs in this field were not investigated.

Author Response

Dear Reviewer 3, Thank you very much for your review. We have addressed the areas you identified and believe the paper has improved and certainly the abstract is better suited to describe the study. Thank you again.

Page

Table

Line(s)

Reviewer Point

Author Response

Edit(s) in Manuscript

In my opinion, the   article would have even more scientific soundness, if there were some   references to the theory of consumption, particularly to the consumer   behavior.

Thank you for your   thoughtful comment. The Health Belief Model and the Health Claims Framework   have both  been used to predict   consumption of health-related products, but we believe that our results are   relevant to a number theories which seek to explain consumption behaviors,   and we hope that researchers will use the findings of our study to further   advance and refine these theories. Unfortunately, however, it isn’t possible   to elaborate on these connections within the limited number of words   allocated to submissions for this journal.

The contribution would   gain in value, if the importance of results for the market would be wider   described. I recommend to relate the study to market investigation or carry   out more interdisciplinary research in future.

Thank you, we will further   collaborate with researchers in other disciplines to expand the application   of the results. We have reduced the length of this manuscript several times   and so for this reason, we will seek collaboration in the future, as you   suggested.

As an aside, our group   published a paper last year that described and characterized the older green   tea consumer in the US which may be used as a consumer profile for QHCs.

We have added language   to suggest that this work be expanded to other disciplines.  

Lines 484-7: “Future   research could examine the complexities of QHCs through a more   interdisciplinary lens. More specifically, collaborating with market   researchers could expand the application of our results to general consumer   behavior and consider its place within the theory of consumption.”

The abstract lacks the   aim of the paper as well as research questions and/or hypotheses.

Thank you for pointing   this out. We have edited the abstract to include the aim and general RQs.

Abstract: Qualified   Health Claims describe diet-disease relationships and summarize the quality   and strength of evidence for a claim. Companies assert that QHCs increase   sales and take legal action to ensure claims reflect their interests. Yet,   there is no empirical evidence that QHCs influence consumers. Using green tea   as a case study, this study investigates the effects of QHCs on purchase   intentions among adults 55 years and older living in the US. An online survey   using a between-subjects design examined QHCs about the relationship between   green tea and the reduced risk of breast and/or prostate cancer or yukichi   fruit juice and the reduced risk of gastrocoridalis, a fictitious   relationship. QHCs written by a green tea company generated greater perceptions   of evidence for the relationship, greater confidence in green tea and cancer   and increased purchase intentions for green tea than other QHCs. Factors that   mitigated the claim’s effects on purchase intentions are: race/ethnicity;   age; importance of health claims; supplement use; health; worry about   health/becoming sick with cancer; worry that led to dietary change; green tea   consumption; and familiarity with the green tea-cancer. Consumers who made   health-related dietary change in the past year and consider health claims   important indicated greater purchase intentions than others.

15

424-6

I think that the   sentence “Perhaps participants in our study that take (….) drink green tea”   (p. 15, l. 424-426) should not appear in the paper because the participants’   beliefs in this field were not investigated.

Thank you for pointing   this out. We have removed this sentence from the manuscript.

Reviewer 4 Report

Qualified Health Claim Language affects Purchase Intentions for Green Tea Products in the United States

The manuscript addresses an interesting topic and fits the scope of the journal Nutrients. Overall I feel this manuscript is strong. The rationale is well articulated and the literature is presented clearly. The study is novel because it researches the effect of Qualified Health Claims on purchase intention. By examining the actual wording of health claims and showing its great impact on consumers, this study is beneficial for marketers and policy makers alike.

The manuscript would benefit from a few minor revisions listed below:

Page 2, Line 54: Incomplete sentence: “Consumer research about Qualified Health Claims.”

Page 6: Please explain why the scale of the variable “Familiarity – diet-disease relationship” is -1, 0, 3.

Page 7 and 8: It should be made clear that the variable “nutrition knowledge” (“knowledge about diet and health”) is subjective nutrition knowledge (or perceived nutrition knowledge like it was written in line 283) because it does not a measure actual knowledge but what the participant perceives to know. See Apendix B in Moorman, C., Diehl, K., Brinberg, D., & Kidwell, B. (2004). Subjective knowledge, search locations, and consumer choice. Journal of Consumer Research, 31(3), 673–680. https://doi.org/10.1086/425102.

The analysis (ind. variables explained almost 40%variance in purchase intentions) is convincing. Based on the good sampling and the shown analysis, the conclusions are justified.

Thank you.

Author Response

Dear Reviewer 4, Thank you very much for your review and attention to detail. I have addressed the issues you identified in the manuscript and believe they improve its quality considerably. Please see the table below for a detailed response to each concern you raised.

Page

Table

Line(s)

Reviewer Point

Author Response

Edit(s) in Manuscript

2

54

“Consumer research about   Qualified Health Claims.”

Thank you for pointing   this out. This is my error and is a subheading. We have changed the   formatting.

Lines 54-5: 1.1. Consumer research about Qualified   Health Claims.

6

2

Please explain why the   scale of the variable “Familiarity – diet-disease relationship” is -1, 0, 3.

Thank you, this is my   error. There are two familiarity variables and the names were mistakenly   swapped. We have changed the variable names to the correct scales.  

Please see changes to the   Familiarity variables in Table 2.

7

283

It should be made clear   that the variable “nutrition knowledge” (“knowledge about diet and health”)   is subjective nutrition knowledge (or perceived nutrition knowledge like it   was written in line 283) because it does not a measure actual knowledge but   what the participant perceives to know. See Apendix B in Moorman, C., Diehl,   K., Brinberg, D., & Kidwell, B. (2004). Subjective knowledge, search   locations, and consumer choice. Journal of Consumer Research, 31(3), 673–680.   https://doi.org/10.1086/425102.

Thank you for noting   this issue. We have changed the language in Table 2 and throughout the   manuscript to address this issue.

Line 235: The majority   of the participants believe they are informed about diet and health (i.e.   perceived nutrition knowledge) (n=1,298, 97.2%) and consider health claims   important on food and dietary supplement product labels (n=1,168, 88.0%;   n=926, 91.5%, respectively).

Lines 237-8: No   differences were found between groups or conditions with respect to general   health, perceived nutrition knowledge, green tea consumption in the past   year, worry about cancer, or health-related dietary changes.

Table 2.: Perceived   Nutrition Knowledge

The analysis (ind.   variables explained almost 40%variance in purchase intentions) is convincing.   Based on the good sampling and the shown analysis, the conclusions are   justified.

Thank you!

Round  2

Reviewer 2 Report

Thanks to the authors for considering and incorporating many of my comments.  Nevertheless, there remain several issues, some more significant than others, that warrant further consideration.

line 67: what are the "four" claims?

Table 1: suggest (1) removing any mention of the fictitious drink and diet-health relationship and (2) adjusting the N to the green tea sample only because none of the fictitious claims existed and could have been "petitioned, unlawfully used or prescribed."

Table 1: should add "scientifically accurate" to the three FDA claims.

Table 1: should substitute "allowable" for "allowed" at the Federal Court claim.

(Significant issue) lines 194-199: strongly suggest that the text clearly indicate the 13-point scale is developed for the purpose of the study but not a scale that FDA has used in its review of scientific evidence or a scale "consistent with" FDA's scoring scheme in any shape or form.

(Significant issue) Lines 229-230: strongly suggest a reconsideration of the logic behind this statement.  this statement appears to mean that only when respondents scored an average of 8 or higher they perceived "greater levels of evidence than was denoted for green tea QHCs by the FDA."  this, however, would preclude the possibility that an average of any number, perhaps except for 1 or 2, could have the same meaning.  In addition, Murphy's ranking approach has not been reviewed and/or accepted by the FDA or any other researchers and should not be used to justify the scale designation here. 

Line 297: 13-point?

(Significant issue) Line 298: the authors' response still has not provided any evidence or established appropriate rationale behind the assertion that "the results are CONSISTENT WITH FDA's evaluation of the scientific evidence."

(Significant issue) Choice of the reference claim: A key research objective is (Lines 140-141) - "whether the current QHC increases consumer purchase intentions for green tea products in comparison with claims no longer in use."  Strongly suggest that the choice of the reference claim be made on this basis.  Also, this choice would be much less vulnerable to the problem of selecting a QHC that covers only one but not both cancers.  And the 2012 claim had about the same number of respondents as other claims.

Author Response

Dear Reviewer 2, 

Thank you for your thoughtful review. We have responded to your concerns and have tabulated them below. Thank you again.

 Page

Table

Line(s)

Reviewer Point

Author Response

Edit(s) in Manuscript

67

what are the "four" claims?

We were referencing the line before (“QHCs composed by the US   FDA and the court in Table 1).

We have clarified the language to address your concern.

Lines 66-7: “For examples of disclaimers, please see the QHCs   composed by the US FDA and the court in Table 1. The disclaimer is the last   sentence of the QHC.”

1

suggest (1) removing any mention of the fictitious drink and diet-health   relationship and (2) adjusting the N to the green tea sample only because   none of the fictitious claims existed and could have been   "petitioned, unlawfully used or prescribed."

(1) We have removed mention of the fictitious condition in this   table.

(2) We respectfully disagree that we halve the n to reflect half of our study sample.   The authors agree that there is value for readers to view the number of   respondents randomly assigned to each QHC group, regardless of the condition.  

Please refer to Table 1.

1

should add "scientifically accurate" to the three   FDA claims.

Ok.

Please refer to Table 1 under column Status.

1

should substitute "allowable" for   "allowed" at the Federal Court claim.

Ok.

Please refer to Table 1 under column Status.

194-199

strongly suggest that the text clearly indicate the   13-point scale is developed for the purpose of the study but not a scale that   FDA has used in its review of scientific evidence or a scale "consistent   with" FDA's scoring scheme in any shape or form.

Ok. We’ve added language to address your concern.

We considered using the FDA QHC language to accurately reflect   the level of evidence in the survey scale. However, the authors strongly agreed   that respondents would likely match words found in the QHC to words found in   the scale, which was not the aim of our study.

Lines 217-9: “The scale does not necessarily reflect FDA’s   rating of evidence however it is based on the Murphy (2005) approach to develop   a scale that may correspond to the levels of evidence [46].”

229-230

strongly suggest a reconsideration of the logic behind this   statement.  this statement appears to mean that only when respondents   scored an average of 8 or higher they perceived "greater levels of   evidence than was denoted for green tea QHCs by the FDA."  this,   however, would preclude the possibility that an average of any number,   perhaps except for 1 or 2, could have the same meaning.  In addition,   Murphy's ranking approach has not been reviewed and/or accepted by the   FDA or any other researchers and should not be used to justify the scale   designation here. 

Ok. We have removed the sentence to address your concern.

REMOVED: The logic followed that scores above the mid-point 7   would align with greater levels of evidence than was denoted for green tea   QHCs by the FDA.

297

13-point?

Ok.

Line 317: 13-point scale of evidence

298

the authors' response still has not provided any evidence or   established appropriate rationale behind the assertion that "the   results are CONSISTENT WITH FDA's evaluation of the scientific   evidence."

Ok. We have edited the language to address your concern.

Lines 316-8: “Overall, participants rated the level of evidence   as a “2” (i.e. D-grade) in both conditions (Table 3) and when the 13-point   scale of evidence was collapsed into four grades of evidence, the results are   consistent with a D level of evidence [54].”

140-141

Choice of the reference claim: A key research objective is   (Lines 140-141) - "whether the current QHC increases consumer purchase   intentions for green tea products in comparison with claims no longer in   use."  Strongly suggest that the choice of the reference claim be   made on this basis.  Also, this choice would be much less   vulnerable to the problem of selecting a QHC that covers only one but not   both cancers.  And the 2012 claim had about the same number of   respondents as other claims.

Thank you. The authors discussed the reference group extensively   during data analysis and revisited the topic in response to your comment. We respectfully   disagree.

The QHC group variables were entered as a block so, no matter   the reference, the result would be similar (Fields) meaning that consumers have   greater intentions when viewing Fleminger QHCs. The ANOVA results under   section 3.1. (Lines 262-4) also demonstrates: “There was also a significant   main effect of QHC, such that claims written by Fleminger produced greater   purchase intentions than claims written by the FDA, F(6, 1,299) =   8.047, < .0001."